# Molecular and Cellular Mechanisms of Intramuscular Fat Development and Growth in Cattle

**DOI:** 10.3390/ijms25052520

**Published:** 2024-02-21

**Authors:** Zhendong Tan, Honglin Jiang

**Affiliations:** School of Animal Sciences, Virginia Tech, Blacksburg, VA 24061, USA; zdtan93@vt.edu

**Keywords:** intramuscular fat, skeletal muscle, beef, nutrition, hormone, epigenetic

## Abstract

Intramuscular fat, also referred to as marbling fat, is the white fat deposited within skeletal muscle tissue. The content of intramuscular fat in the skeletal muscle, particularly the *longissimus dorsi* muscle, of cattle is a critical determinant of beef quality and value. In this review, we summarize the process of intramuscular fat development and growth, the factors that affect this process, and the molecular and epigenetic mechanisms that mediate this process in cattle. Compared to other species, cattle have a remarkable ability to accumulate intramuscular fat, partly attributed to the abundance of sources of fatty acids for synthesizing triglycerides. Compared to other adipose depots such as subcutaneous fat, intramuscular fat develops later and grows more slowly. The commitment and differentiation of adipose precursor cells into adipocytes as well as the maturation of adipocytes are crucial steps in intramuscular fat development and growth in cattle. Each of these steps is controlled by various factors, underscoring the complexity of the regulatory network governing adipogenesis in the skeletal muscle. These factors include genetics, epigenetics, nutrition (including maternal nutrition), rumen microbiome, vitamins, hormones, weaning age, slaughter age, slaughter weight, and stress. Many of these factors seem to affect intramuscular fat deposition through the transcriptional or epigenetic regulation of genes directly involved in the development and growth of intramuscular fat. A better understanding of the molecular and cellular mechanisms by which intramuscular fat develops and grows in cattle will help us develop more effective strategies to optimize intramuscular fat deposition in cattle, thereby maximizing the quality and value of beef meat.

## 1. Introduction

Meat products have historically served as a substantial source of protein, essential amino acids, vitamins, and trace minerals for the global population. While the sale of beef is restricted in certain countries due to religion and culture, it consistently remains one of the most consumed meat products among the general populace. Consumers generally perceive health benefits as derivable from adequate beef consumption compared to other protein sources [1,2]. A mere 55 g or 45 g of high-quality beef can satisfy the daily protein requirements of an adult male or female, respectively [3]. With the rise in global prosperity, the demand for high-quality beef has continuously increased.

Besides protein, beef contains a significant amount of fat. Intramuscular fat (**IMF**) is defined as the fat situated between bundles of muscle fibers [4]. Physiologically, IMF is believed to have similar functions to other fat depots in cattle, serving as an energy reserve and providing fuel during times of increased metabolic demand or inadequate nutrient supply. As food, IMF substantially enhances the texture and flavor of the meat and the overall satisfaction of consumers [5]. The marketing of fresh beef relies significantly on IMF, especially in the *longissimus dorsi* muscle (**LM**), where higher grades command higher prices. The current dietary trend suggests that consumers are more concerned about saturated fatty acids (**SFA**) in the diet, as a high intake of SFA may increase the risk of cardiovascular disease [6]. Intramuscular fat contains more monounsaturated (**MUFA**) and polyunsaturated fatty acids (**PUFA**) than other fat depots in cattle [3]. Beef with a higher percentage of IMF, or a higher degree of “marbling”, has a higher ratio of MUFA to SFA, and is therefore considered healthier meat [7].

The development of IMF in cattle accelerates post-puberty and is influenced by various factors including genetics, nutrition, sex, and management practices [8]. This means that the development of IMF has great potential for intervention [9]. Nonetheless, our understanding of the molecular and cellular mechanisms governing the growth and development of IMF in cattle and other meat animals remains limited. In this review, we summarize the process of IMF development and growth in cattle and discuss the major factors that affect this process and the molecular and epigenetic mechanisms that mediate IMF development and growth in cattle.

## 2. Types of Adipose Tissue in Cattle

Adipose tissue can be categorized into white and brown adipose tissue based on its physiological function and color. Brown adipose tissue (**BAT**) is characterized by the presence of multiple small lipid droplets, a high mitochondrial density, and its association with non-shivering thermogenesis, primarily facilitated by the activity of uncoupling protein 1 (**UCP1**) [10]. In a newborn calf, approximately 2% of its body weight is BAT, which is primarily distributed around the kidneys and under the skin [11]. Bone morphogenetic protein 7 (**BMP7**) promotes the differentiation of brown preadipocyte primarily by activating a complete program of brown adipogenesis, including the induction of early regulators of brown fat fate such as PR domain containing 16 (**PRDM16**) and peroxisome proliferator-activated receptor-gamma coactivator-1alpha (**PPARGC1A**), the increased expression of *UCP1* and adipogenic transcription factors such as peroxisome proliferator-activated receptor-gamma (***PPARG***) and CCAAT/enhancer-binding proteins (***CEBPs***), and the induction of mitochondrial biogenesis through the p38 mitogen-activated protein kinase and PGC-1-dependent pathways [12]. This specialized fat tissue serves as a vital source of heat generation, ensuring the newborn calves can effectively regulate their body temperature and thrive in their early stages of life. When faced with low temperatures, the mortality rate of Brahman calves is higher than that of Angus calves [13]. This difference can be attributed to several factors: (1) Angus calves have a higher quantity of BAT, with a greater cell density; (2) the mitochondria in Brahman BAT tend to be spherical, whereas they are more elongated in Angus BAT, and the cross-sectional area of mitochondria is often larger in Brahman than Angus BAT; (3) under cold exposure, lipogenesis is more prevalent in Angus than in Brahman BAT [14,15]. As animals age, brown adipose tissue gradually transforms into white adipose tissue due to altered hormone levels, with specific BMPs such as BMP2 and BMP4 contributing to the enhanced white adipogenesis [16,17].

White adipose tissue (**WAT**) is characterized by its adipocytes containing a single large lipid droplet [18]. White adipose tissue plays a critical role in storing energy and regulating the overall balance of fatty acids in the body. When there is an excess of calories, WAT stores fatty acids as triglycerides through lipogenesis; when animals are in a fasting state, WAT releases fatty acids from stored triglycerides through lipolysis [19]. In humans and rodents, the liver is the main site for the de novo synthesis of fatty acids; in ruminants, fat and muscle tissue are the main sites for this synthesis [20]. Additionally, in ruminants, both acetate and glucose are the major carbon sources of fatty acid synthesis, whereas in nonruminants, glucose is the major carbon source of fatty acid synthesis [20]. Because acetate is a major product of rumen fermentation, ruminants such as cattle are more prone to depositing IMF than monogastric animals such as pigs.

Adipose tissue can be further classified into subcutaneous adipose tissue, visceral adipose tissue, bone marrow adipose tissue, intermuscular adipose tissue, and intramuscular adipose tissue, based on its location in the body. In cattle, IMF differs from other white fat depots not only in location but also in biological function, and, more notably, economic value. Because IMF presents as fat deposited within the muscles, it can directly influence the energy balance and growth of muscle tissue [21]. Intramuscular fat in cattle enhances the flavor and juiciness of meat; as such, it is considered “valuable” fat. In contrast, subcutaneous fat and other white fat deposits in cattle are considered “waste” fat in the meat industry, as they take a lot of dietary energy to grow yet are avoided by most consumers, because eating too much fat might increase the risk of cardiovascular disease [22]. 

## 3. Development and Growth of Intramuscular Fat in Cattle

### 3.1. Adipogenesis

Adipocytes are terminally differentiated cells of adipose tissue, and the change of adipose tissue mass depends on both the hyperplasia (cell number increase) and hypertrophy (cell size increase) of adipocytes [23]. Adipocytes originate from mesenchymal stem cells (**MSCs**) or adipose progenitor cells (**APCs**) [24]. Adipogenesis can be divided into two stages: commitment (or determination) and differentiation [25] (Figure 1). The transition from APCs or MSCs to preadipocytes, which is marked by the expression of the *PPARG* gene, is controlled at the transcriptional level by several transcription factors and extracellular signals (Figure 1). Bone morphogenetic proteins (**BMPs**) [16,26], platelet-derived growth factor receptor alpha (**PDGFRA**) [27], and zinc finger proteins 423 and 467 (**ZFP423** and **ZFP467**) [28] are the major regulators of the commitment of MSCs or APCs to preadipocytes, while factors like RUNX1 partner transcriptional co-repressor 1 (**RUNX1T1**) [29] suppress this commitment [30].

The process of adipogenic differentiation has been extensively studied, with PPARG and CEBPs identified as the core regulators of this process [31]. The *CEBPB* and *CEBPD* genes are expressed early in adipogenic differentiation, and they induce the co-expression of *PPARG* and *CEBPA*, two crucial transcription factors in the later stages of adipogenesis [32,33,34]. These transcription factors ultimately activate the expression of genes specific for adipocytes and the deposition of lipids, resulting in the formation of mature adipocytes [35]. The process of differentiation of preadipocytes into adipocytes is also controlled by hormones and other extracellular signals such as leptin and testosterone (Figure 1), and these extracellular signals will be discussed in detail in the sections below. 

### 3.2. Adipogenesis within Skeletal Msucle

The majority of research on the developmental biology of IMF has been conducted in the mouse model, although mice are not known for their ability to accumulate IMF. In mice, IMF formation starts in the late stages of pregnancy [36]. In the process of embryonic skeletal muscle development, primary myofibers are first formed followed by secondary myofibers [37]. The formation of adipocytes, i.e., adipogenesis, in skeletal muscle overlaps that of secondary myofibers in the middle or late stages of pregnancy [38]. It is highly likely that mesenchymal progenitor cells differentiate into either myogenic or fibro/adipogenic lineage [20]. Cells from the myogenic lineage further develop into muscle fibers, intramuscular brown adipocytes, and satellite cells, while cells from the fibro/adipogenic lineage develop into white adipocytes and fibroblasts [39]. Since myogenic and adipogenic lineage cells originate from the same pool of stem cells, the commitment of stem cells to myogenesis or adipogenesis is a competing process. More myogenesis implies suppressed adipogenesis, and vice versa. The formation of initial adipocytes has a dominant effect on the number of total intramuscular adipocytes. As a result of the paracrine effects and their close proximity to adipocytes, skeletal muscle insulin resistance develops, causing a shift in the commitment of stem cells from myogenesis to adipogenesis [40,41]. 

During embryonic development, the Wnt signaling pathway activates myogenic and osteogenic differentiation while inhibiting the adipogenic differentiation of mesenchymal multipotent cells [38,42]. This primarily involves Wnt signaling suppressing the key adipogenic regulators CEBPA and PPARG [43]. Farmer and colleagues observed a functional interaction between β-catenin (a component of the Wnt signaling pathway) and PPARG, wherein these two proteins negatively regulate the activity of each other [44]. The Wnt signaling pathway is thought to be involved in maintaining a balance between adipogenesis and myogenesis. Specifically, the loss of WNT10B, either due to aging or targeted gene deletion, leads to an increased adipogenic potential of myoblasts and the acquisition of adipocyte characteristics during the process of muscle regeneration [45]. Bromodomain-containing protein 4 (**BRD4**) is a member of the bromodomain and extra-terminal domain (**BET**) family of proteins, which are epigenetic readers that recognize acetylated histones and facilitate the recruitment of transcriptional regulators that promote the expression of *PPARG*. *BRD4* knockout inhibits the expression of *PPARG* and suppresses adipogenesis [46]. Cooperating with lineage-determining transcription factors (**LDTFs**), BRD4 recruits regulators such as MLL3/MLL4 and CBP/p300 to the enhancers [47]. *BRD4* knockout mice showed reduced brown fat and muscle mass, indicating that BRD4 is an important factor for both adipogenesis and myogenesis [47]. Myostatin (**MSTN**) is a major negative regulator of skeletal muscle development and growth [48]. Myostatin also inhibits adipose tissue growth by reducing lipid accumulation through the ERK1/2 and PKA signaling pathways [49]. In addition to multipotent stem cells, highly committed cells like myoblasts [50], fibroblasts [51], and pericytes [52] in muscle tissue can also be induced to differentiate into mature adipocytes at least in vitro, as depicted in Figure 2.

In cattle, embryonic stem cells begin to form adipocytes at three months of gestation. Visceral adipocytes are formed first, followed by subcutaneous adipocytes, intermuscular adipocytes, and lastly intramuscular adipocytes [53]. The first intramuscular adipocytes are generated at around 180 days of gestation [54]. Since intramuscular fat matures later than subcutaneous fat, pursuing a high level of marbling often increases the overall carcass fat, particularly subcutaneous fat [55]. Fortunately, the hyperplasia phase of intramuscular adipocytes continues until 250 days of age, far surpassing other fat cell populations, which stop hyperplasia during the weaning stage (intermuscular and subcutaneous adipocytes) or the neonatal stage (visceral adipocytes) [56]. The differential timing of adipocyte hyperplasia creates the concept of a “marbling window” (from 150 to 250 days of age), which provides an opportunity for stimulating intramuscular adipocyte formation without increasing overall fat accumulation. During this period of marbling window, active intramuscular preadipocyte proliferation is a major mechanism of IMF deposition [57]. From 250 days of age to slaughter, the hypertrophy of adipocytes plays a more important role than hyperplasia in IMF growth [53,58]. 

The primary driving force in current research on IMF is to find a way to improve IMF growth while avoiding an overall increase in body fat. A number of genes that potentially regulate IMF development and growth in cattle have been identified (Table 1). These genes are often the focus of research aimed at evaluating how various factors, such as dietary supplements, influence adipose tissue development and growth in cattle.

## 4. Factors That Determine Intramuscular Fat Development and Growth in Cattle and the Underlying Mechanisms

The deposition of intramuscular fat in cattle is influenced by various factors, as shown in Figure 3.

### 4.1. Breed

Undoubtedly, breed is a major factor affecting the IMF content of beef. Different cattle breeds have different growth rates and fat distribution characteristics, all of which may affect their IMF content. Wagyu cattle exhibit a remarkable capability for IMF deposition, with 37.8% of the mass of LM being IMF, while Brahman cattle have a notoriously low capability for IMF deposition, with IMF representing only 2.8% of LM [78]. The breeds currently widely used in beef cattle production such as Angus and Hereford are fast-growing and muscular, but have a relatively low IMF content in the longissimus muscle (around 10%) [79,80]. Some local breeds such as Hungarian Grey also show the same ability to deposit intramuscular fat as Angus [81].

### 4.2. Sex

The impact of sex on IMF deposition in cattle is notable. Bulls exhibit stronger muscle development but weaker fat accumulation abilities than cows and steers. Park and associates compared the IMF content in bulls, castrated bulls, and cows, and found that castrated bulls had the highest marbling scores, followed by cows [82]. Because the impact of male sex hormones on fat accumulation is significant (discussed in more detail below), castration is a common strategy employed to enhance IMF deposition in bulls, although this reduces the growth rate [83]. The castration of young bulls reduced the expression of the *WNT10B* and *CTNNB1* genes but increased the expression of the Wnt antagonist *SFRP4* and adipogenesis-related genes *CEBPA* and *PPARG* [75]. Correlation analyses revealed a negative association of the IMF content with the levels of *WNT10B* and *CTNNB1* mRNAs, and a positive correlation with the levels of *SFRP4*, *CEBPA*, and *PPARG* mRNAs. These data suggest that testosterone, the major male sex hormone in bulls, inhibits IMF deposition by decreasing the expression of genes such as *CEBPA* that mediate adipogenesis and increasing the expression of genes such as *WNT10B* that inhibit adipogenesis. Sex also has a significant effect on fatty acid composition. Compared to bulls, heifers have higher levels of monounsaturated fatty acids in IMF, and these differences are likely attributed to the differential expression of genes such as acyl-CoA desaturase (***SCD***), fatty acid synthase (***FASN***), and growth hormone (***GH***) in the IMF between the two sexes [84,85].

### 4.3. Weaning Age

Weaning is the stage at which calves transition from being entirely dependent on mother’s milk to consuming solid feed. This stage significantly influences the physiological and developmental aspects of calves. Proper weaning contributes to the improved utilization of feed, promoting growth and weight gain [86]. It is currently believed that early weaning, especially when combined with supplemental feeding, can enhance IMF deposition. Tipton and colleagues found that early weaning at around 150 days of age combined with isonitrogenous rumen bypass lipid supplementation significantly increased the IMF content in Angus steers, without negatively affecting the dressing percentage, when compared to traditional weaning at around 210 days of age [87]. Angus–Wagyu crossbred heifers subjected to early weaning (142 days of age) and a high-concentrate diet showed greater IMF deposition and weight gain compared to heifers weaned at 180 days and grown on pasture for 16 months before entering the feedlot [88]. Angus × Gelbvieh and Angus steers subjected to earlier weaning (90 days of age) exhibited higher marbling scores compared to those weaned traditionally (174 days of age) [89]. These data suggest that early weaning combined with high-concentrate feeding before grazing or feedlot finishing is a viable strategy to enhance IMF deposition.

A study was conducted to explore the potential mechanisms by which early weaning increases IMF deposition in Angus and Angus × Simmental steers [61]. The study revealed that early weaning and a high-starch diet activated the key fat synthesis regulatory factors, *PPARG* and *CEBPA*, promoting the early maturation of preadipocytes and fat accumulation. In another study, cattle were fed different diets for 10 weeks at 2 weeks of age, including mother’s milk + roughage (control group), milk replacer + concentrate, milk replacer + concentrate + roughage, and milk replacer + concentrate + 30% starch, and after this all calves were fed the same solid diet for 22 months. Compared to the control group, the group fed milk replacer + concentrate + roughage showed changes in the PPAR signaling pathway, the cytoskeletal regulation of actin, unsaturated fatty acid biosynthesis, and the Wnt signaling pathway. The group fed milk replacer + concentrate + 30% starch exhibited changes in the insulin and mTOR signaling pathways. Changes in the adiponectin signaling pathway occurred in both the milk replacer + concentrate + roughage group and the milk replacer + concentrate group compared to the control group [90]. As cattle weight increases, the contribution of glucose to fat synthesis decreases, while acetate utilization increases, especially in the intramuscular fat [7]. Thus, earlier weaning combined with concentrate feeding introduces glucose to calves earlier, consequently increasing the deposition of fat in muscle tissues.

### 4.4. Slaughter Age and Weight

Intramuscular fat deposition increases with age in most breeds of cattle. For instance, in Holstein-Friesian × Hereford bull cattle, the IMF content in LM at 15 months of age was 15.9%, and this percentage increased to 18.3% at 18 months of age [91]. The IMF content in LM of Japanese Black steers cloned from somatic cells increased with the slaughter age too: 23.7% at 20 months, 38.7% at 25 months, and 41.1% at 30 months of age [92]. The IMF contents and marbling scores of five different muscle groups (biceps femoris, supraspinatus, semitendinosus, longissimus lumborum, and infraspinatus) in three common beef cattle breeds (Angus, Hereford, and Wagyu × Angus) increased with the slaughter age [93]. Extending the slaughter age is an effective strategy to enhance IMF deposition in cattle. However, with the increase in animal age, feed efficiency significantly decreases, leading to increased feeding costs, increased overall carcass fat, and decreased tenderness of meat [94]. In various cattle breeds, the IMF content and marbling score increase with the slaughter weight too. Data from British and Japanese Black × Holstein crossbred cattle showed a linear increase in the IMF content as carcass weight increased from 200 to 400 kg during a fattening period [95]. Similarly, Angus steers exhibited a linear increase in IMF accumulation as the slaughter weight increased from 208 to 380 kg [96]. As animals grow, although the rate of muscle growth decreases, the rate of fat gain remains constant, and this explains why the IMF content usually increases with slaughter weight [97]. 

### 4.5. Nutrition

#### 4.5.1. Energy Level

Animals require extra energy for the accumulation of intramuscular fat. Grains, as energy-dense feed, can be efficiently utilized by the rumen and small intestine to produce acetate and glucose, substrates for fat synthesis. Steers fed a high-energy diet had a higher IMF content compared to those fed a low-energy diet [98]. Kong and colleagues fed iso-nitrogenous diets with different energy levels (low 3.72 MJ/kg, medium 4.52 MJ/kg, and high 5.32 MJ/kg) to three groups of yaks for 120 days, and found that animals fed the high-energy diet accumulated significantly more IMF than the other two groups, accompanied by the greater expression of fatty acid synthase (***FAS***), acetyl-CoA carboxylase (***ACACA***), sterol regulatory element-binding protein 1 (***SREBF1***), *SCD*, lipoprotein lipase (***LPL***), fatty acid-binding protein 3 (***FABP3***), and *PPARG* mRNAs in IMF [99]. In exploring the impacts of dietary energy levels on the rumen microbiome and the fatty acid profile in the muscles of fattened yaks, subsequent studies found that increasing dietary energy increased the ratio of Firmicutes to Bacteroidetes and stimulated the relative abundance of populations such as *Streptococcus bovis*, *Prevotella ruminicola*, and *Ruminobacter amylophilus*, and that Prevotella was positively correlated with the intramuscular content of total polyunsaturated fatty acids and negatively correlated with the intramuscular content of total saturated fatty acids [99]. These results suggest that dietary energy level might affect IMF deposition or composition through the modulation of the rumen microbial community.

Triglyceride synthesis relies on the availability of the supply of fatty acids, whether synthesized de novo or obtained from the diet [95,100]. The pathways involved in de novo fatty acid synthesis, as depicted in Figure 4, show how high-energy diets benefit IMF deposition. Acetate and citrate serve as substrates for fatty acid synthesis, with acetate primarily originating from rumen fermentation and citrate produced through glycolysis and the tricarboxylic acid cycle. Additionally, glucose can be generated via the gluconeogenesis of propionate and lactate or absorbed from small intestinal digestion [101]. Two key enzymes in the pathway of de novo fatty acid synthesis using glucose are ATP citrate lyase and NADP malic enzyme. Studies have indicated that ATP citrate lyase exhibits lower activity in ruminants, resulting in less utilization of glucose as a substrate for fat synthesis in comparison to monogastric animals [102]. However, glucose is considered a more suitable substrate than acetate for fatty acid synthesis during IMF deposition [19,103]. Furthermore, IMF displays a higher expression of glucose transporter 4 (***FABP4***), which facilitates glucose uptake in adipose tissue, than subcutaneous fat [104]. Smith and colleagues reported that acetate accounted for 70–80% of acetyl units in subcutaneous fat but only 10–25% in IMF, and that, in contrast, glucose contributed to 1–10% of acetyl units in subcutaneous fat but 50–75% in IMF [103]. Similar findings were presented by Rhoades and colleagues [105]. In addition to fatty acids, the synthesis of triglycerides needs glycerol, which is primarily sourced from glucose (Figure 4). Grain-fed cattle have easier access to glucose and faster IMF deposition compared to pasture-fed cattle [106]. Therefore, providing sufficient sources of glucose to cattle during the fattening period helps to increase the IMF content [95]. Research has indicated that cattle fed high-concentrate diets display an elevated expression of fat synthesis transcription factors (*CEBPB*, *CEBPA*, and *PPARG*) in IMF [107]. Thus, high-energy concentrates may help activate key transcription factors that promote triglyceride synthesis and, consequently, intramuscular fat accumulation.

#### 4.5.2. Dietary Fats

Due to the presence of high microbial activity in the rumen, dietary fats undergo significant modifications before reaching the small intestine for absorption. Within the rumen, dietary fats undergo hydrolysis, releasing fatty acids that are subject to extensive biohydrogenation by the microbial community. This process results in the loss of approximately 90% of dietary polyunsaturated fatty acids (e.g., linoleic acid), while the amount of saturated fatty acids (e.g., stearic acid) significantly increases [108]. Rumen microbes also synthesize certain lipids from complex plant-based polysaccharides like cellulose, pectin, and starch, leading to a surplus of lipids within this compartment compared to the initial intake [108]. Unesterified fatty acids are not absorbed in the rumen but rather adhere to particulate matter and enter the abomasum in this form [109]. In the abomasum, the rumen microbes are broken down, releasing lipids. These lipids, along with the fatty acids present in the ingested food, subsequently enter the small intestine. The efficient regulation of the digestion and absorption of dietary fats in the small intestine can enhance the deposition of IMF in cattle. For instance, supplementing emulsifiers to steers during the early or late stages of the fattening phase improved carcass marbling scores [110,111]. Furthermore, bile acids, which are crucial components of bile, play an important role in the digestion and absorption of lipids while maintaining lipid metabolism balance. Bile acids bind to lipid molecules in the small intestine, facilitating the intestinal digestion and absorption of lipids. The supplementation of cholic acid, a bile acid, has been found to improve IMF deposition and meat quality in heifers [112]. 

#### 4.5.3. Vitamins A and D

Vitamins are a class of organic compounds that are essential in small amounts for growth, development, immunity, metabolism, and many other physiological processes [113]. Vitamins cannot usually be synthesized by the body and need to be obtained from foods or supplements. Among all vitamins, vitamins A and D have profound effects on the development and function of adipocytes.

Vitamin A is a fat-soluble vitamin crucial for maintaining normal growth and development, immune system function, vision, skin health, and reproduction [114]. Many studies have shown that vitamin A affects IMF deposition in cattle. For instance, in Wagyu [115], Holstein [116], and Angus cattle [117], vitamin A, either as a dietary supplement or in the form of injections, is inversely correlated with the IMF content. Research conducted by Kruk and colleagues showed that restricting the intake of vitamin A in Angus steers promoted the deposition of IMF, primarily through the increased hyperplasia of adipocytes [118]. Vitamin A reduces the synthesis of fatty acids and insulin sensitivity in skeletal muscle [119]. When the concentration of vitamin A in IMF decreases due to the absence of supplemental vitamin A in the feed, the extractable fat contents in the muscle or marbling scores are enhanced [120].

The metabolite of vitamin A, retinoic acid (**RA**), strongly inhibits the early differentiation of preadipocytes. Its receptors (**RARA**, **RARB**, and **RARG**) can form heterodimers with retinoid X receptor (**RXR**) or the transcription factor PPARG to regulate gene transcription [121]. The main biologically active forms of RA, 11-cis retinal and all-trans retinoic acid (**ATRA**), suppress the expression of *CEBPA*, *PPARG*, and their target genes [122,123]. Retinoic acid induces the expression of preadipocyte factor 1 (***DLK1***), *KLF2*, and sex-determining region Y box 9 (***SOX9***), which are imprinting markers of preadipocytes, by activating cellular RA binding protein II (**CRABP-II**) and RARG. The RA inhibition of *CEBPA* and *PPARG* expression in preadipocytes inhibits the differentiation of preadipocytes into adipocytes, and maintains the morphology and phenotype of preadipocytes. Meanwhile, RA regulates the expression of *FABP5* and *PPARA*, promoting lipid oxidation and energy expenditure in mature adipocytes [124]. Therefore, limiting the amount of vitamin A in the diet during the late-finishing phase (20–24 month of age) of cattle production may improve IMF deposition in cattle. However, the continued restriction of vitamin A is not advisable, as this may lead to health issues and slower growth rates [115]. 

Interestingly, vitamin A has a positive effect on intramuscular fat deposition during the early life (before 250 days of age) of cattle. Felipe and colleagues (2022) found that a one-shot injection of 300,000 IU of vitamin A into the muscles of Montana × Nellore calves at birth increased IMF deposition [125]. Injecting vitamin A (150,000 IU) to neonatal and one-month-old Angus steers increased the number of intramuscular PDGFRα-positive adipose progenitor cells, enhanced the fat-generating potential of intramuscular SVF cells, and upregulated the expression of vascular endothelial growth factor A (***VEGFA***) [126]. These results suggest that the positive effect of vitamin A on the intramuscular adipogenic capacity in young calves may be related to the enhanced angiogenesis in muscle.

Vitamin D is a fat-soluble vitamin too, and is best known for its critical role in bone health and absorption and the metabolism of calcium and phosphorus. Vitamin D has been shown to suppress the differentiation of preadipocytes into adipocytes and the expression of genes associated with fat production [127,128]. Vitamin D acts by impeding key factors responsible for lipogenesis, notably PPARG, thereby reducing lipid accumulation [129]. The bioactive form of vitamin D, 1,25-dihydroxyvitamin D3 (calcitriol), exerts its effects on gene expression by binding to the vitamin D receptor (**VDR**), which forms a heterodimer with the RXR. When it comes to the influence of vitamin D on IMF deposition in cattle, there have been limited in vivo studies conducted. One study showed that vitamin D supplementation reduced body fat content in Angus-crossbred steers [130]. Thus, limiting vitamin D intake during the fattening phase (20–24 month of age) might help improve IMF deposition in cattle. However, vitamin D intake should not be limited during the growth phase of cattle because Vitamin D is essential for bone growth. 

### 4.6. Hormones

#### 4.6.1. Leptin

Leptin is a protein hormone primarily produced by white adipose tissue. This hormone is mainly involved in regulating body weight and energy balance. It accomplishes these functions by transmitting lipostatic signals from adipocytes to leptin receptors in the hypothalamus, leading to appetite suppression and increased thermogenesis [131,132]. Research has shown a positive correlation between plasma leptin concentrations and the lipid content of LM in Wagyu cattle [133]. Polymorphisms in the leptin gene have been associated with serum leptin levels and body fat content [131,134]. Daix and colleagues found that Belgian Blue cattle had the lowest body fat and IMF, as well as the lowest serum leptin concentrations and leptin gene expression in adipose tissue, when compared to Limousin and Angus cattle [135]. Therefore, serum leptin concentration may be used as a biomarker for the IMF content in cattle (Figure 1).

#### 4.6.2. Thyroid Hormones

Thyroid hormones are a group of hormones produced by the thyroid gland that are crucial for maintaining normal physiological functions and metabolism [136]. Thyroid hormones play various roles in fat metabolism, increasing the basal metabolism to promote energy expenditure and facilitating the oxidation of fatty acids rather than their storage as fat [137]. However, Mears and associates believe that thyroid hormones, specifically thyroxine (**T4**) and triiodothyronine (**T3**), may play a role in enhancing IMF deposition in cattle, although the corresponding molecular mechanisms have not yet been discovered [138]. The participants in the thyroid hormone signaling pathway are significantly correlated with IMF in cattle. For example, the mRNA expression of the thyroid hormone-responsive protein (***THRSP***) gene, which is regulated by thyroid hormones and primarily expressed in adipose tissue [139], is often higher in muscles with higher IMF compared to those with lower IMF [140]. *THRSP* is mainly expressed in mature adipocytes rather than in the early stages of fat generation [141]. Additionally, the thyroglobulin (***TG***) gene, which encodes a precursor to thyroid hormones, has polymorphisms that directly impact cattle IMF deposition [142]. All these data suggest a role for thyroid hormones in regulating IMF deposition in cattle.

#### 4.6.3. Sex Steroids

Steers and heifers exhibit greater IMF deposition capabilities than bulls [143]. These differences suggest that sex hormones play a role in IMF deposition in cattle. Male sex steroid hormones testosterone and dihydrotestosterone stimulate muscle growth but suppress fat deposition in cattle. Singh and colleagues found that testosterone promoted the commitment of the MSC cell line C3H10T1/2 into the myogenic lineage, but inhibited their determination to become fat cells [144]. Testosterone inhibited the differentiation of IMF preadipocytes from cattle into adipocytes in culture by suppressing the activity of glycerol-3-phosphate dehydrogenase (**GAPDH**), a lipogenesis-related enzyme [145]. On the other hand, two female sex steroid hormones, 17β-estradiol and progesterone, enhanced GAPDH activity and suppressed the expression of lipolysis-related genes such as *LPL*. Additionally, progesterone upregulated the expression of the *SREBF1* gene [146]. These data provide potential mechanisms by which testosterone inhibits IMF deposition, whereas estrogen and progesterone stimulate IMF deposition in cattle.

### 4.7. Stress

Stress is often associated with depressed IMF deposition. Like inadequate energy intake or disease, stress can lead to a surge in blood cortisol levels and adipose tissue breakdown [147,148]. In cattle, cold stress triggers fat reserve breakdown, redirecting energy towards heat production to help maintain body temperature [149]. Heat stress adversely affects growth and fat deposition in cattle, particularly impacting IMF [150]. In situations of heat stress, the levels of hormones leptin and adiponectin are upregulated, suppressing animal feed intake and fat deposition [151]. Although there is evidence indicating that heat stress reduces the activity of lipolytic enzymes in adipose tissue, inhibiting fat oxidation and causing a decrease in circulating non-esterified fatty acids, there is a simultaneous increase in the activity of adipose tissue lipoprotein lipase, and this elevation increases the degradation of circulating triglycerides in the bloodstream [152].

### 4.8. Epigenetic Control of IMF Development and Growth in Cattle

Epigenetic changes are changes in gene expression that do not stem from alterations in DNA sequences. Epigenetic mechanisms that alter gene expression patterns primarily involve DNA methylation and histone modifications, such as histone acetylation and methylation, leading to changes in DNA–protein interaction, chromatin architecture, as well as DNA-non-coding RNA interaction [153]. By regulating gene expression, epigenetic mechanisms influence intramuscular fat development and deposition (Figure 5).

#### 4.8.1. DNA Methylation

DNA methylation stands as one of the predominant epigenetic modifications, known for its relative stability and pivotal role in gene regulation [154]. DNA methylation occurs primarily at the 5′ position of DNA within the cytosine loop adjacent to guanine, also referred to as the cytosine–phosphor–guanine (CpG) sites or islands [155]. DNA methyltransferases (DNMTs) facilitate the methylation of cytosine at the 5′ position, yielding 5-methylcytosine (5-mC). In the promoter region, CpG island methylation directly interferes with transcription by obstructing the binding of transcription factors [156]. DNA methylation can undergo changes when organisms encounter specific stimuli, and variations in DNA methylation levels and patterns may serve as significant factors in governing tissue-specific gene expression and cellular differentiation [157].

The status of DNA methylation can regulate gene expression in IMF, thereby influencing IMF deposition. For instance, Fang and colleagues conducted a comprehensive whole-genome DNA methylation analysis using whole-genome bisulfite sequencing (**WGBS**) on two cattle breeds, Wagyu and Chinese Red Steppes cattle, which differ significantly in the content of IMF. They identified 23,150 differentially methylated regions within 8596 genes and 1046 significantly enriched gene ontology (**GO**) terms, including those related to lipid transport and transportation [158]. DNA methylation is known to play an important role in the transcriptional activation of key regulators, such as PPARG, in fat production [159]. Consequently, DNA methylation could be one of the mechanisms governing gene expression associated with adipocyte formation and lipid synthesis in IMF.

Baik and associates found that the methylation levels of two CpG sites within the *PPARG* promoter region and five CpG sites within the *FABP4* promoter region were lower in IMF than in surrounding muscle [160]. Consequently, the mRNA levels of *PPARG* and *FABP4* genes were higher in IMF than in muscle LM [160]. Huang and colleagues isolated and immortalized SVF cells from the Sternocleidomastoid muscle of Angus heifer calves, and found that clones with higher adipogenic capacity displayed significantly lower DNA methylation levels in the *ZFP423* promoter region compared to those with lower adipogenic capacity [70]. The reduced DNA methylation levels in IMF are linked to the decreased expression of genes encoding methyltransferases, like *DNMT3A* and *DNMT3B*, which are the pivotal methyltransferases in DNA methylation. In the offspring of Wagyu-crossbred cows, the expression levels of *DNMT3A* and *DNMT3B* were notably lower in muscle than in liver, and there was a correlation between the expression levels of *DNMT3A* and *DNMT3B* in the liver tissue and the marbling score in beef cattle [161]. 

#### 4.8.2. Histone Modifications

DNA base pairs tightly wrap around histones to form a DNA–protein complex known as the nucleosome. Chromatin can be visualized as a chain of nucleosomes within somatic cells, and its condensation or unraveling depends on nuclear signals [162]. There are four primary histones, H2A, H2B, H3, and H4, which compose the octameric core of the nucleosome, and H1, which binds to DNA and stabilizes the nucleosome structure [163]. Various reversible post-translational modifications to amino acids on the N-terminal end of histone tails can modulate the conformation and packaging of these histones, determining DNA accessibility for transcription.

Histone modifications influence cell development and fate by interacting with gene promoters and enhancers, and thereby modulating gene transcription [164,165,166,167]. Little research has been conducted to determine the effects of histone modifications on IMF development in cattle. The acetylation of lysine 4 of histone 3 protein (**H3K4ac**), a histone mark often associated with transcriptional activation or open chromatin conformation, is the direct target of sirtuin 1 (**SIRT1**), a deacetylase. SIRT1 suppresses the differentiation of IMF adipose precursor cells into adipocytes by reducing H3K4ac at three potential new transcription factors (***NRF1***, ***NKX3-1***, and ***EGR1***) and four other genes (***MAPK1***, ***RXRA***, ***AGPAT1***, and ***HADH***) involved in the endoplasmic reticulum and MAPK signaling pathways [168]. These data suggest a role for histone modification in regulating gene expression in IMF, and hence IMF development. 

#### 4.8.3. Small Non-Coding RNAs

In epigenetics, non-coding RNAs (**ncRNAs**) play a role in controlling gene expression, silencing transcriptional elements, inactivating X chromosomes, alternative splicing, and DNA imprinting [169]. Among ncRNAs, microRNAs (**miRNAs**) are the most extensively studied; miRNAs typically consist of 20 to 30 nucleotides, are highly conserved among species, and make up approximately 1 to 1.5% of all cellular transcripts in animals [170]. Mature miRNAs guide the RNA-induced silencing complex (**RISC**) to bind to complementary sequences in the 3′ untranslated region (**UTR**) of messenger RNAs (**mRNAs**), and this binding can lead to translational repression and/or mRNA cleavage [171]. MiRNAs play a fundamental role in various biological processes, including cell proliferation, differentiation, and apoptosis [172].

Multiple studies suggest that miRNAs play a role in adipocyte differentiation, lipid synthesis, and cholesterol metabolism [173,174,175]. For example, during the differentiation of mouse mesenchymal stem cells into adipocytes, the expression of 66 miRNAs changed, suggesting the involvement of specific miRNAs in the expression of genes related to adipogenesis [176]. Similarly, differences in miRNA expression patterns have been observed between subcutaneous fat and IMF in adult cattle. MiRNAs such as miR-143, miR-145, miR-26a, miR-2373-5p, and miR-23b-3p were expressed at higher levels in IMF, whereas miR-26a, miR-2373-5p, miR-2325c, miR-3613, and miR-2361 were more abundant in subcutaneous fat [177]. Functional enrichment analyses of genes predicted to be targeted by these differentially expressed miRNAs have suggested that miRNAs may control adipogenesis by controlling the expression of genes associated with the mitogen-activated protein kinase, Wnt, and transforming growth factor-β signaling pathways. Overexpressing miR-143 in preadipocytes isolated from steers’ IMF enhanced the differentiation of preadipocytes into adipocytes. Conversely, reducing miR-143 expression inhibited adipogenic differentiation and promoted the proliferation of preadipocytes [178]. Chen et al. found that the transfection of Qinchuan cattle intramuscular preadipocytes with bta-miR-376a mimic inhibited the mRNA and protein levels of *CDK1*, *CDK2*, *PCNA*, *CEBPA*, *FAS*, and *PPARG*, as well as cell proliferation. The dual-luciferase reporter system identified that bta-miR-376a targeted *KLF15*, a key transcription factor in adipogenesis [179]. Similarly, Seong et al. discovered that bta-miR-494 was negatively correlated with marbling score in Korean cattle, and that the downregulation of bta-miR-494 was associated with the upregulation of the *KLF11* gene [180]. These data suggest that miRNA-143, bta-miR-376a, bta-miR-494, and likely many other miRNAs are involved in the deposition of IMF in cattle.

In addition to miRNAs, other types of non-coding RNAs may be involved in the development of intramuscular adipocytes too. For example, using comparative transcriptomics and competitive endogenous RNA (**ceRNA**) network analysis of mRNAs and regulatory RNAs associated with IMF and fat metabolism in the muscle tissue of five cattle breeds (Angus, Chinese Simmental, Luxi, Nanyang, and Shandong Black), Reyhan and colleagues identified 34 circular RNAs (**circRNAs**) and 57 long non-coding RNAs (**lncRNAs**) in addition to 15 miRNAs and 374 mRNAs that may participate in IMF deposition and metabolism in cattle [181]. 

Clearly, the possible roles of the aforementioned small non-coding RNAs in IMF deposition need to be further investigated with in vivo studies. 

#### 4.8.4. Maternal Effect

Epigenetic markers are capable of being passed down from one cell to another during lineage development, and when acquired early in life, they can influence the adult phenotype. Furthermore, these markers have the potential to impact the phenotype of subsequent generations [182,183]. The transgenerational inheritance of epigenetic modifications has been a challenging concept to elucidate, as epigenetic modification patterns are typically reset in gametes following fertilization [184,185]. However, increasing evidence suggests that certain epigenetic markers can “escape” this resetting [186]. 

In sheep, maternal obesity during pregnancy was associated with an increased lipid content in the muscle of their offspring at 22 months of age [187]. Additionally, nutrient restriction from day 28 to 78 of pregnancy elevated the concentration of intramuscular triglycerides in the muscle of offspring at 8 months of age [188]. In cows, protein restriction during pregnancy resulted in a reduction in early collagen content in calf skeletal muscle and insufficient collagen remodeling in later stages [189]. Excessive nutrition during mid-pregnancy impacted fat deposition, altered the expression of fat deposition markers (*CTNNB1*, *PPARG*, *ZNF423*), and tended to increase the expression of fiber deposition markers (*COL1*) in fetal skeletal muscle [190]. Increasing the feed intake of pregnant Nellore cows to 1.5 times the maintenance requirement elevated the mRNA expression of *ZFP423*, *CEBPA*, and *PPARG,* which, as mentioned above, are key transcriptional regulators of adipogenesis, in their offspring’s skeletal muscle; however, there were no significant changes in the expression of myogenic-related genes *MYOD1* and *MYOG* [191]. These data indicate that maternal overnutrition enhanced the formation of intramuscular fat in the offspring. Feeding cows with methyl donor-rich diets before and during early pregnancy altered the methylation and transcriptome expression patterns in the muscle of their offspring [192]. The offspring of beef cows fed hay during pregnancy had higher marbling scores and IMF deposition compared to those of beef cows fed corn during pregnancy [193]. Wang and colleagues discovered differences in the expression of imprinted genes *H19*, *MEG8*, *IGF2R*, and *DNMT3A* in the muscle between offspring of cows fed a high-starch and those of cows fed a low-starch diet during pregnancy [194]. Moisá and others found that the expression of miRNAs, particularly miR-34a, which is involved in anti-fat generation, in calf muscle was influenced by maternal nutrition [195]. These data suggest that IMF deposition in calves may be improved by manipulating the maternal diets. These data also suggest that the transgenerational effects of maternal diets during pregnancy on intramuscular fat development in the offspring may be mediated through the transgenerational transmission of diet-caused epigenetic changes. 

## 5. Conclusions and Future Directions

Intramuscular fat has a significant impact on the quality and value of beef. The development of IMF in cattle requires an adequate number of committed adipose precursor cells. The differentiation of these precursor cells to adipocytes and the deposition of lipids in adipocytes are subject to the regulation of many factors. Practical measures such as high-energy feeding, castration, early weaning, and extended finishing before slaughter can modulate the differentiation of adipocyte precursor cells and the expression of genes related to this process, thereby increasing the development and deposition of IMF. Furthermore, nutrition can impact the expression of adipogenesis-related genes through epigenetic modifications, and epigenetic modifications may even be transmitted to the offspring and thus have an impact on the development of IMF in the offspring. Although significant progress has been made in understanding these dynamics, much remains to be understood about the molecular mechanisms underlying the development and growth of IMF in cattle. Future research should take advantage of innovative technologies such as epigenomics, proteomics, metabolomics, meta-genomics, and nutrigenomics to explore the intricate interplay among the various factors, and thus glean a deeper comprehension of the mechanisms of IMF development and growth in cattle and meet the surging demand for high-quality meat while safe-guarding the sustainability of cattle production.

## Figures and Tables

**Figure 1 ijms-25-02520-f001:**
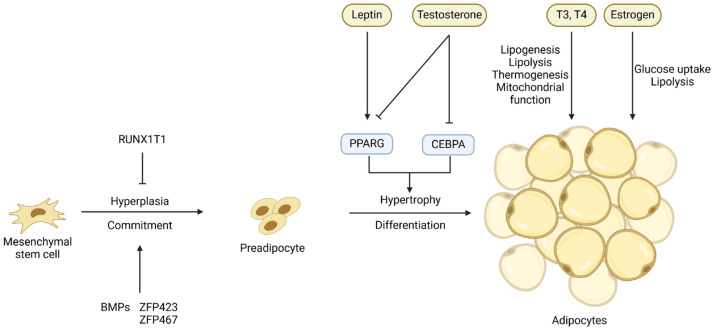
Key transcriptional and hormonal regulators of the commitment of mesenchymal stem cells or adipose progenitor cells to preadipocytes and the differentiation of preadipocytes into adipocytes. Arrows denote stimulation and T-shaped lines denote inhibition.

**Figure 2 ijms-25-02520-f002:**
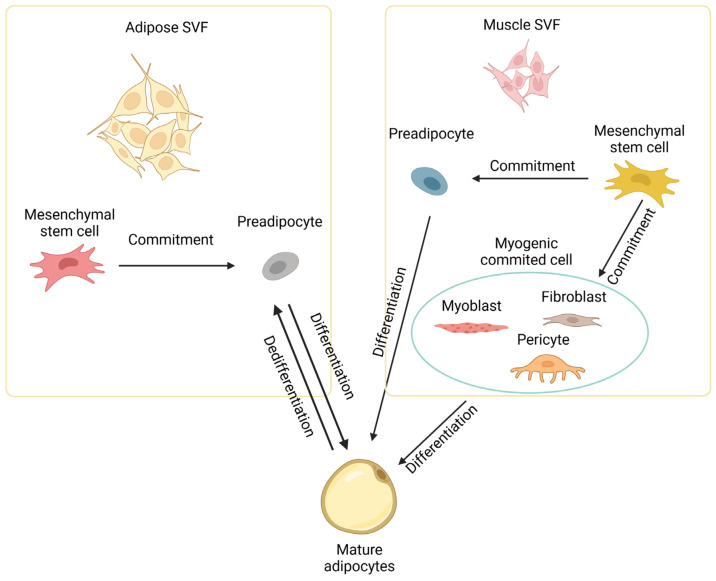
Development of intramuscular adipocytes. Intramuscular adipocytes can be formed from mesenchymal stem cells in the stromal vascular fraction (SVF) of both adipose and muscle tissues through commitment and differentiation. Intramuscular adipocytes might also be formed from committed cells like myoblasts, fibroblasts, and pericytes in muscle. Diagram has been drawn based mainly on mouse studies. See text for details.

**Figure 3 ijms-25-02520-f003:**
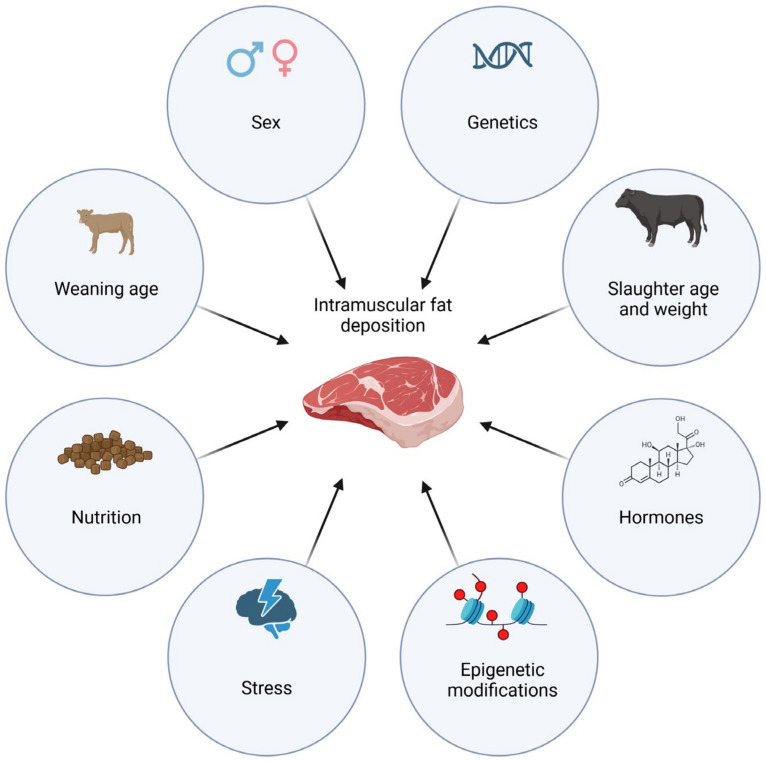
Factors affecting intramuscular fat deposition in cattle.

**Figure 4 ijms-25-02520-f004:**
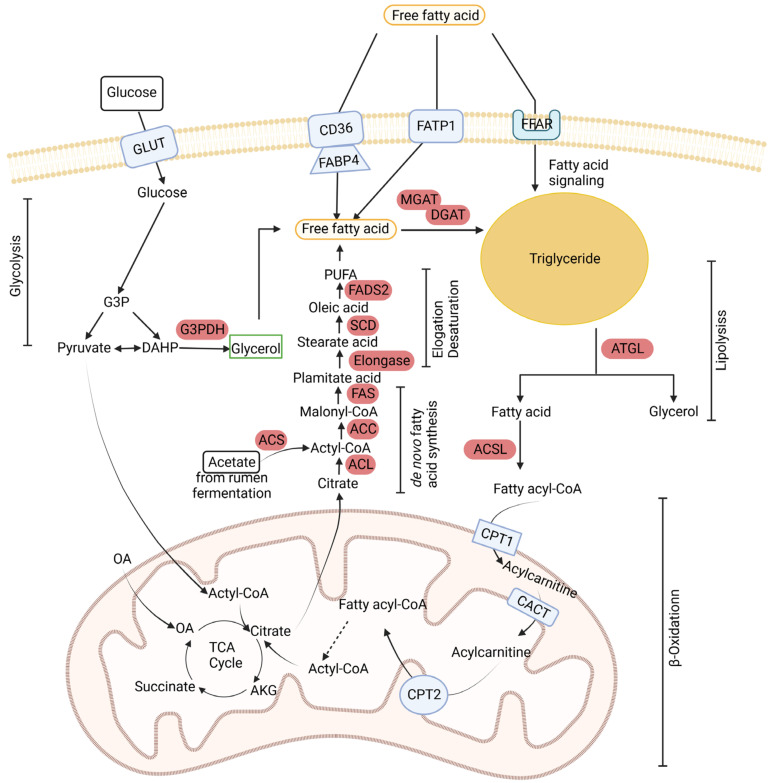
A schematic diagram of bovine metabolic pathways involving glycolysis, fatty acid and triglyceride synthesis, and fatty acid beta-oxidation. GLUT: Glucose transporter. CD36: Fatty acid translocase. FABP1 and 4: Fatty acid-binding protein 1 and 4. FFAR: Free fatty acid receptor. CPT1 and 2: Carnitine palmitoyltransferase 1 and 2. CATC: Carnitine acylcarnitine translocase. G3P: Glyceraldehyde-3-Phosphate. OA: Oxaloacetate. AKG: α-Ketoglutarate. G3PDH: Glycerol-3-phosphate dehydrogenase. ACS = ACSL: Acyl-CoA synthetase. ACL: ATP-citrate lyase. ACC: Acetyl-CoA carboxylase. FAS: Fatty acid synthase. SCD: Stearoyl-CoA desaturase. FADS2: Fatty acid desaturase 2. MGAT: Monoglyceride acyltransferase. DGAT: Diacylglycerol O-acyltransferase. ATGL: Adipose triglyceride lipase.

**Figure 5 ijms-25-02520-f005:**
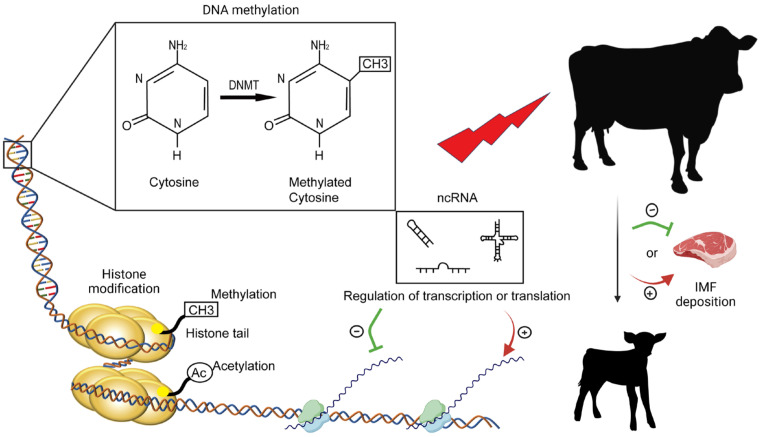
Three epigenetic mechanisms—DNA methylation, histone modification, and non-coding RNA-binding—mediate the effects of various factors, such as maternal nutrition, on intramuscular fat deposition in cattle.

**Table 1 ijms-25-02520-t001:** Genes associated with intramuscular fat development in cattle.

Genes	Species	Association with IMF	Impact	References
*ICER*	Hanwoo	Highly expressed in the late fattening stage and during preadipocyte differentiation into adipocytes	+	[59]
*WISP2*	Wagyu	Highly expressed in adipose precursor cells	+	[60]
*PPARG*		A master transcriptional regulator of adipose differentiation	+	[61]
*PDGFRA*	Angus	Abundance is correlated with IMF content	+	[62]
*CEBPA*		A core transcription factor of fat differentiation	+	[63]
*SCD1*	Wagyu, Simmental	Increases IMF deposition and unsaturated fatty acid content	+	[64]
*KLF*	Yak	Inhibitory factor of IMF differentiation	−	[65]
*DGAT1*	Holstein, Charolais,	Involved in fatty acid esterification and correlated with IMF content	+	[66,67,68]
*ADIPOQ, THRSP*	Wagyu × Hereford	Abundance is highly correlated with IMF content	+	[69]
*ZFP423*	Angus	Promotes adipogenic differentiation of muscle stromal vascular cells	+	[70]
*ACSL1*	Charolais × Holstein	Regulates lipid composition and polyunsaturated fatty acids synthesis in adipocytes	+	[71,72]
*CD36*	Qinchuan	Cattle with the combined genotype WWCCAA in the CD36 gene had higher IMF contents	+	[73]
*ABHD5*	Qinchuan	An accelerator for adipose triglyceride lipase	−	[74]
*WNT gene family*	Korean	Inhibits the differentiation of fat cells	−	[75]
*FOXO1*	Luxi	Affects the expression of genes associated with apoptosis of intramuscular adipocytes	−	[76]
*KLF3*	Qinchuan	The polymorphism of the KLF3 gene has an impact on the intramuscular fat content	−	[77]

ICER: Inducible cAMP early repressor. WISP2: WNT1 inducible signaling pathway protein 2. PPARG: Peroxisome proliferator-activated receptor gamma. PDGFRA: Platelet-derived growth factor receptor alpha. CEBPA: CCAAT/enhancer-binding protein alpha. SCD1: Stearoyl-CoA desaturase 1. KLF: Krüppel-like factor. DGAT1: Diacylglycerol O-acyltransferase 1. ADIPOQ: Adiponectin. THRSP: Thyroid hormone responsive. ZFP423: Zinc finger protein 423. ACSL1: Acyl-CoA synthetase long-chain family member 1. CD36: Cluster of differentiation 36. FOXO1: Forkhead box O1. ABHD5: Abhydrolase domain containing 5. The “+” and “−“ signs indicate positive and negative effects of the genes on IMF deposition, respectively.

## Data Availability

Not applicable.

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
