# Peer review of "Molecular and Cellular Mechanisms of Intramuscular Fat Development and Growth in Cattle"

_ijms, 2024, doi:10.3390/ijms25052520_

Round 1

Reviewer 1 Report

Comments and Suggestions for Authors

This manuscript presents an overview of the regulatory mechanisms from both in vivo and in vitro perspectives to elucidate intramuscular fat development and growth in cattle. However, several areas require improvement. Below are suggestions for a major review:

Line 19-21: The conclusion lacks final remarks. Adding summarizing statements would enhance clarity and impact.

Line 90-99: The transition from adipose progenitor cells to preadipocytes should be elaborated in a distinct section for better coherence and understanding.

Line 91-107: The discussions surrounding preadipocyte and adipocyte mechanisms predominantly reflect outdated perspectives. Authors are encouraged to integrate recent studies that offer compelling insights.

Line 109: Given the extensive mention of signaling pathways and mechanisms pertinent to myogenesis and myoblast development, it is recommended that the section title be revised for greater precision.

Line 110-138: The paper highlights β-catenin and Wnt10b as key genes in adipogenesis and myogenesis. However, exploration of more recent genes is necessary, as the current discussion does not significantly diverge from existing literature.

Line 139: The term "Redifferentiation" used in Figure 1 and the categorization of progenitor cells warrant a more cautious approach due to ongoing debates in the field.

Line 144-157: This section, hypothesizing the transition from preadipocyte to adipocyte development, should be supported by actual in vivo data from cattle globally.

Line 164: The impact of specific functional genes on intramuscular fat in cattle, whether positive or negative, needs to be clearly delineated.

Line 174-183: The exclusive mention of Wagyu beef in the context of marbled beef production overlooks other breeds like Angus and Korean native cattle, among others, suggesting a need for a broader perspective.

Line 185-198: The discussion appears limited to a few traditional genes and signaling pathways, suggesting the necessity for a more expansive exploration of genetic factors.

Line 199: The section on weaning age focuses narrowly on select cattle breeds, failing to represent the diversity of breeds capable of achieving high marbling scores.

Line 329-354: The paper omits significant details on the role of vitamin A in cattle, including its influence on preadipocyte development in postnatal calves and its effects on marbling scores across cattle breeds. The implications of oral vitamin A supplementation also require clarification.

Line 329-364: While discussing vitamins and vitamin A, the absence of information on receptor interaction (e.g., RAR, RXR, VDR) is a notable gap.

Line 484-518: The section on non-coding RNAs lacks empirical support, highlighting a need for in vivo studies to validate the claims made (except for a few simple screenings in the paper).

Line 523-550: This segment should be integrated with earlier discussions on cattle, as the current focus on traditional pathways does not sufficiently distinguish this paper from prior work.

Reviewer 2 Report

Comments and Suggestions for Authors

General comment

I read this interesting manuscript. It fits the scope of the journal and I endorse acceptance for publication after addressing the appended comments and questions.

Line 21: please add a highlight statement about the content of this review. For example: In this review study, we extrapolated diverse factors that control the intramuscular fat in cattle……..etc.

Line 35: please add one or two sentences about the biological roles of intramuscular fat in cattle.

Line 45-46: please state some of these factors.

Line 47-51: Please consider these lines to address my first comment.

Line 62: factors not fact-ors (a typing error). Please address this issue in the whole manuscript

Line 66-67: please explain the reason.

Line 79-81: please edit it again.

Line 84-88: please add a scientific explanation.

Line 90-108: I recommend an illustrating diagram to visualize the process of adipogenesis. What about the age of animals?

Line 140-143: For this figure, please indicate whether this illustration is derived from mice studies or cattle ones.

Line 172-173: an illustrating figure could be helpful for the readership, please.

Line 177: It would be better to add an illustrating figure showing the breed effect and the morphology of intramuscular fat, especially for marbling meat of Wagyu cattle.

Line 185-198: The names of genes should be in italics, I think. Please, address this issue in the whole manuscript.

Line 231-232: consequences are what?

Line 275: all citing figures in the text should be bold, I just recommend it!

Line 343-344: please specify this phase (approximate times).

Line 362-362: Comparable to vitamin A, what authors recommend during the early age and finishing phase (before slaughter).

Line 365: for hormonal regulation of the IMF in cattle, I strongly recommend an illustrating figure to enhance understanding of this issue.

Line 396-420: in light of your statements, what are the consequences of castration on IMF deposition and quality?

Also, could the authors recommend the breeders or companies (to invest this information) to castrate bulls or not? and specify the suitable age if you recommend castration.

Line 484-514: This section is very nice. Just enumerate the limitations of research on this topic.

Line 523: this section illustrated the effect of different maternal factors on the IMF parameters of their newborn. What about the effect of these factors on the quality of IMF on the mother itself? For example, the impact of body condition score, pregnancy toxemia, restriction of protein, or undernutrition.

Line 541-543: in light of your statements, several researchers recommend a dietary supplement of energy substrate, antioxidants, and others to combat the negative effect of this critical period on the overall reproductive performance of animals. Could you please add a highlight spot on the consequences or economic importance of these additives on the quality of the IMF on the growing calves?

Final comment: I recommend acceptance of this good manuscript. However, some factors may be not covered well in this draft such as the effect of transportation before the animal slaughter. Also, I wonder whether the method of animal slaughter could impact the quality of the IMF or not. Collectively, this manuscript is valuable.

Reviewer 3 Report

Comments and Suggestions for Authors

The authors collected all the scientific information about cattle's intramuscular fat development process. It is a straightforward reading work which is well understandable and deeply scientific. The reviewer believes no more subchapters could be involved in the whole story of fat in cattle from unborn foetal brown fat through different types of white fat tissues in the animals until finally reaching the highly requested well-marbled (high intramuscular containing) beef.

It is an easy decision.

The text, subchapters, and figures are a coherent complex altogether.

Maybe one more table could be helpful for the readers to digest these huge data and various biochemical pathways, which included in fat deposition process related to ages (maternal-foetal, newborn, before weaning, weaning to slaughter age.

The authors referred 168 scientific works, which is quite a high number with several mistyping, incorrect citations in references. It should be corrected before the acceptance of the manuscript.

Round 2

Reviewer 1 Report

Comments and Suggestions for Authors

This manuscript is suitable for publication after these modifications. 

Reviewer 2 Report

Comments and Suggestions for Authors

The manuscript was improved and i recommend ACCEPTANCE.